# Exopolysaccharides Producing Bacteria for the Amelioration of Drought Stress in Wheat

**Noshin Ilyas** [1,*], **Komal Mumtaz** [1], **Nosheen Akhtar** [1], **Humaira Yasmin** [2], **R. Z. Sayyed** [3], **Wajiha Khan** [4], **Hesham A. El Enshasy** [5,6,7,*], **Daniel J. Dailin** [5,6], **Elsayed A. Elsayed** [8,9] and **Zeshan Ali** [10]

1 Department of Botany, PMAS Arid Agriculture University, Rawalpindi 46300, Pakistan; komalmumtaz070@gmail.com (K.M.); noshee.nawaz444@gmail.com (N.A.)

2 Department of Biosciences, COMSATS University, Islamabad 45550, Pakistan; humaira.yasmin@comsats.edu.pk

3 Department of Microbiology, PSGVP Mandal's Arts, Science and Commerce College, Shahada 425409, Maharashtra, India; sayyedrz@gmail.com

4 Department of Biotechnology, COMSATS University Islamabad, Abbottabad Campus, Abbottabad 22010, Pakistan; wajihak@cuiatd.edu.pk

5 Institute of Bioproduct Development (IBD), Universiti Teknologi Malaysia (UTM), Skudai 81310, Johor, Malaysia; jddaniel@utm.my

6 School of Chemical and Energy Engineering, Faculty of Engineering, Universiti Teknologi Malaysia (UTM), Skudai 81310, Johor, Malaysia

7 City of Scientific Research and Technology Applications (SRTA), New Burg Al Arab 21934, Alexandria, Egypt

8 Zoology Department, College of Science, King Saud University, P.O. 2455, Riyadh 11451, Saudi Arabia; eaelsayed@ksu.edu.sa

9 Chemistry of Natural and Microbial Products Department, National Research Centre, Dokki, Cairo 11651, Egypt

10 Plant Physiology Program, Crop Sciences Institute, National Agricultural Research Centre, Park Road, Islamabad 45500, Pakistan; eco4nd@yahoo.com

* Correspondence: noshinilyas@yahoo.com (N.I.); henshasy@ibd.utm.my (H.A.E.E.)

**Abstract:** This research was designed to elucidate the role of exopolysaccharides (EPS) producing bacterial strains for the amelioration of drought stress in wheat. Bacterial strains were isolated from a farmer's field in the arid region of Pakistan. Out of 24 isolated stains, two bacterial strains, *Bacillus subtilis* (Accession No. MT742976) and *Azospirillum brasilense* (Accession No. MT742977) were selected, based on their ability to produce EPS and withstand drought stress. Both bacterial strains produced a good amount of EPS and osmolytes and exhibited drought tolerance individually, however, a combination of these strains produced higher amounts of EPS (sugar 6976 µg/g, 731.5 µg/g protein, and 1.1 mg/g uronic acid) and osmolytes (proline 4.4 µg/mg and sugar 79 µg/mg) and significantly changed the level of stress-induced phytohormones (61%, 49% and 30% decrease in Indole Acetic Acid (IAA), Gibberellic Acid (GA), and Cytokinin (CK)) respectively under stress, but an increase of 27.3% in Abscisic acid (ABA) concentration was observed. When inoculated, the combination of these strains improved seed germination, seedling vigor index, and promptness index by 18.2%, 23.7%, and 61.5% respectively under osmotic stress (20% polyethylene glycol, PEG6000). They also promoted plant growth in a pot experiment with an increase of 42.9%, 29.8%, and 33.7% in shoot length, root length, and leaf area, respectively. Physiological attributes of plants were also improved by bacterial inoculation showing an increase of 39.8%, 61.5%, and 45% in chlorophyll a, chlorophyll b, and carotenoid content respectively, as compared to control. Inoculations of bacterial strains also increased the production of osmolytes such asproline, amino acid, sugar, and protein by 30%, 23%, 68%, and 21.7% respectively. Co-inoculation of these strains enhanced the production of antioxidant enzymes such as superoxide dismutase (SOD) by 35.1%, catalase (CAT) by 77.4%, and peroxidase (POD) by 40.7%. Findings of the present research demonstrated that EPS, osmolyte, stress hormones,

and antioxidant enzyme-producing bacterial strains impart drought tolerance in wheat and improve its growth, morphological attributes, physiological parameters, osmolytes production, and increase antioxidant enzymes.

**Keywords:** antioxidant enzymes; *Azospirillum brasilense*; *Bacillus subtilis*; stress hormones; osmolytes; plant growth promotion; wheat

## 1. Introduction

Wheat is a staple food used worldwide and its demand will increase with the growing population. It is assumed that its production needs to be increased by 50% by the year 2030 [1]. Wheat and other crop plants are continuously exposed to various biotic and abiotic stress factors that reduce plant growth and productivity. Abiotic stress factors include heat, salinity, chilling, nutrient deficiency, and drought. Climate change is predicted to increase the severity of extreme weather which may affect crop production worldwide. In the current situation, periodic drought is the main cause of reducing the global productivity of plants including wheat [2]. Drought stress is caused by the limited availability of water in the soil which ultimately reduces turgor, nutrient absorption, growth, and yield of plants. Stomatal closure results in decreased photosynthetic rates in plants. Plants adjust their vital processes to combat drought stress. However, the plant responses are controlled by their genetic makeup and the duration of the drought stress [3].

There is a need to develop and promote environment-friendly strategies to cope with drought stress. Inoculation of plant growth-promoting rhizobacteria (PGPR) in the rhizosphere of crops is one of the most suitable strategies in agriculture [4]. The PGPR is a group of beneficial soil bacteria that not only improve the growth of the plants but also enhance drought tolerance in plants [5]. There are several mechanisms by which PGPR work efficiently under stress. The reported mechanisms include the production of stress-related hormones, osmolytes, antioxidant enzymes; exopolysaccharides (EPS), etc. [6]. Under water stress, the inoculation of EPS producing PGPR strains improves plant growth [7]. Literature reports that EPS changes soil structure and improves root/shoot growth. As a result of the change in the rhizosphere, plants expand their root system which enhances the nutrient and water uptake. It also detoxifies free radicals and reduces the negative effects of reactive oxygen species (ROS) [8]. EPS-producing strains have been also reported to produce antioxidant enzymes i.e., superoxide dismutase (SOD), catalase (CAT), and peroxidase (POD), which detoxify ROS, therefore the ability of PGPR to augment the antioxidants can help in imparting drought tolerance. On the other hand, osmotic adjustment is also the key adaptation at the cellular level. It is the accumulation of organic and inorganic solutes at the cellular level which maintain the cells' turgor properties and also protect proteins, enzymes, membranes, and cellular organelles from oxidative damage [9]. Proline is one of the prominent osmolytes produced in drought stress. It scavenges free radicals and stabilizes sub-cellular structures. The PGPR inoculation is known to improve proline levels and antioxidant activities and thus help plants to withstand drought or any other abiotic stress conditions [10].

Though rhizobacterial inoculations have been studied in cereals, however, identification and investigation of the role of EPS producing PGPR strains in combating drought conditions still need in-depth exploration. This research hypothesized that EPS producing PGPR are less prone to drought and their inoculation in the rhizosphere ameliorates the negative impacts of drought stress in wheat. This study elucidates the role of EPS producing PGPR on germination, physiological, morphological, and biochemical parameters of wheat under drought stress.

## 2. Materials and Methods

### 2.1. Collection of Soil Samples and Isolation of Bacteria from Soil

Two soil samples were collected from topsoil (6 inches) of farmer's fields at the arid region of Bahawalpur, (29°25′5.0448″ N–71°40′14.4660″ E) Pakistan. A composite sample was obtained from each location. Soil samples were stored in plastic bags and were brought to the laboratory where they were stored at 4 °C, before further use. The soil samples were analyzed for physico-chemical characteristics by following the method of Li et al. [11].

The serial dilution method was followed for the isolation of bacteria from the two soil samples. One gram of soil sample was suspended in 9 mL of distilled water. This soil suspension was stirred with a magnetic stirrer for 1 h followed by centrifugation (3000 rpm) for 10 min. Supernatant was used to prepare decimal dilutions and the resulting aliquots (20 μL) were spread on Luria-Bertani (LB) agar plates and nitrogen-free bromothymol (NFB) semisolid media of the following composition (g/L); KOH 4.0, DL-Malic acid 5.0, $K_2HPO_4$ 0.5, $MnSO_4.H_2O$ 0.01, $FeSO_4.7H_2O$ 0.05, $MgSO_4.7H_2O$ 0.1, $CaCl_2.2H_2O$ 0.01, NaCl 0.02, $Na_2MoO_4.2H_2O$ 0.002, agar-agar 1.75, bromothymol blue 2 mL (0.5% alcoholic solution) and distilled water 1000 mL. The pH of the medium was adjusted to 6.8 and incubation was done at 28 °C for 2 days. Pure cultures were obtained through streaking the colonies 4–5 times on LB agar by dilution plate technique until a single pure colony appeared [12].

### 2.2. Characterization of Bacterial Isolates

Isolates were identified based on colony morphology (shape, margin, size, elevation, texture, appearance, optical properties, and pigmentation) as well as for cellular characteristics (cell shape, motility, gram staining) [13]. Each of the purified bacterial isolates was evaluated for its plant growth promotion (PGP) characteristics.

The phosphorous solubilization ability of the isolates was checked by their production of cleared zones around the colonies in the Pikovaskaya'sagar. Phosphomolybdate blue color method was used to measure the total solubilized phosphate [14]. Slightly modified Pikovoskaya'smedium was inoculated with each strain followed by incubation at 30 °C for 5 days. After incubation, the broths were centrifuged for 15 min at 6000 rpm. A mixture of the supernatant (500 μL) and 2,4-dinitrophenol (40 μL) was added to 20 μL of dilute sulfuric acid, following the addition of 5 mL of chromogenic reagent. The resulting mixture was diluted using sterilized water to make the total volume up to 50 mL and absorbance (680 nm) was recorded.

Siderophore production was tested according to the modified method of Patel et al. by spot inoculation on Chrome Azurol S (CAS) media [15]. Each of the bacterial strains was spot inoculated on petri plates containing CAS media. An un-inoculated plate was considered as control. The inoculated CAS medium plates were incubated for approximately a week at 28 °C and the colonies thus obtained were observed for the production of the orange zone around the bacterial colonies.

Each of the bacterial isolates was also checked for its ability to produce hydrogen cyanide (HCN) following the method of Lorck [16]. Nutrient agar medium (pre-soaked in 2% sodium carbonate *w/v* and 0.5% picric acid) was streaked with bacterial strains supplemented with 4.4 g/L of glycine. The inoculated plates were sealed with para-film paper and incubated at 28 °C for 4 days. The development of color (orange or red) is an indication of the production of HCN.

A series of conventional biochemical tests were carried out (according to Bergey's Manual of Systematic Bacteriology) to characterize the isolated bacteria [17]. The carbon/nitrogen (C/N) source utilization pattern of bacterial isolates was determined using QTS-24 kits. The isolates were then tested for their drought tolerance by growing each of them in LB agar supplemented with different levels (5%, 10%, 15%, 30%, 25%) of PEG 6000 [16].

PCR-Amplification and 16S rRNA Sequence Analysis

DNA was extracted from pure cultures of each strain grown in LB broth as described by Chen and Kuo [18]. Amplification of genomic DNA of bacterial isolates was carried out according to the method Weisburg et al. [19]. The primer used for PCR-amplification has the nucleotide sequence of (fd1) AGAGTTTGATCCTGGCTCAG, and reverse primer (rd1) (AAGGAGGTGATCCAGCC). The amplified PCR product was purified and sequenced on an automated sequencer by gel purification kits (JET quick, Gel Extraction Spin Kit, GENOMED). The strains were identified by using a nearly complete sequence of 16S rRNA gene on (BLAST) NCBI by comparing sequence and phylogenetic relatedness with the known organisms.

### 2.3. Assessment of Drought Tolerance

Drought tolerance of the two selected bacterial strains was tested by culturing them in a flask containing 50 mL of nutrient broth medium supplemented with 10% and 20% polyethylene glycol (PEG6000). For the combination, the isolates were grown in 100 mL nutrient broth (NB) and were incubated at 28 °C, 150 rpm for 24 h. The cells were separated via centrifugation at 4000 rpm for 15 min, and the pellet was suspended in 10 mL sodium chloride (NaCl; 0.85 g) normal saline for preparation of standard inoculums. The growth of each isolate was monitored by measuring the absorbance at 550 nm by using a spectrophotometer till the optical density 0.5, calculated for both isolates as 10% (*v/v*) of standard inoculate. Then, 10% (*v/v*) of standard inoculate was transferred into 100 mL of the medium with 10% and 20% polyethylene glycol and incubated for 7 days at 28 °C in a shaker at 150 rpm. Their growth rate was analyzed by using the protocol described by Mishra [20]. The concentration of osmolytes was also measured by following the protocol of Hodge and Hofreiter [21].

### 2.4. EPS Emulsification Activity of Bacterial Strains

Bacterial strains were also tested for EPS-emulsification activity by following the protocol described by Rosenberg et al. [22]. Lyophilized EPS (0.5 mg) was dissolved in distilled water (0.5 mL) by heating at 65 °C for ~20 min, later cooled at room temperature. The total volume was made 2 mL by adding phosphate buffer saline (PBS) to the EPS solution. This mixture was supplemented with hexadecane (0.5 mL) and vortexed for 5 min. The absorbance of the resulting solution was taken instantly after vortex ($A_0$) using a spectrophotometer at 540 nm. Another reading of absorbance was taken after incubating the mixture at room temperature for 30 and 60 min ($A_t$). For the emulsifying activity of EPS% retention after incubation time was calculated as;

$$t = A_t/A_0 \times 100 \qquad (1)$$

A control (blank) was prepared as a mixture of hexadecane (0.5 mL) and PBS (2 mL).

### 2.5. Quantification of EPS Substances

Polysaccharide extraction and analysis were done by using the protocol of Naseem and Bano et al. [23]. The bacterial strains were cultured in optimized mineral salts medium with 18.2% $KH_2PO_4$, 12.6% $K_2HPO_4$, 0.6% $MnSO_4$, 10% $NH_4NO_3$, 0.06% $FeSO_4.2H_2O$, 1% $CaCl_2.2H_2O$, 1% $MgSO_4.7H_2O$, 1.5% NaCl, 1% sodium molybdate, and 0.2% of glucose in 1 L of distilled water for 10 days. The bacterial cultures (250 mL) were centrifuged at 15,000 rpm at 4 °C for 20 min. The supernatant was used to extract the EPS by the addition of two-fold ice-cold ethanol (95%). For complete precipitation, the solution was chilled at 4 °C. The EPS was obtained from the chilled solution. The solubility of EPS was determined by suspending small quantities of lyophilized EPS in 2 mL of chloroform, benzene, acetone, water, ethanol, and methanol. Pellet formation was observed after the mixture was vortexed and allowed to stabilize for some time. The concentration of protein, total soluble sugar, and uronic acid was determined by using the protocols of Lowrey et al. [24],

Dubois et al. [25], and Taylor and Buchanan [26], respectively. The medium without inoculum was used as the blank.

### 2.6. Screening and Production of Phytohormones

The ability of two drought-tolerant strains to produce phytohormones such as indole acetic acid (IAA), gibberellic acid (GA), abscisic acid (ABA), and cytokinins (CK) in the culture media was determined according to the method of Tien et al. [27]. The hormone extraction was carried out by centrifugation of the cultured broths at 10,000 rpm for 15 min followed by adjustment of pH (2.8) with 1 N HCl and addition of an equal volume of ethyl acetate. The residue obtained after the evaporation of the solution at 35 °C, was mixed with 1500 μL of methanol. Finally, the samples were run on HPLC (Agilent 1100, Germany) equipped with a C18 column (39 × 300 mm) and a UV detector. For standardization of HPLC, pure grade hormones; CK, ABA, IAA, and GA (Sigma Chemical Co., St. Louis, MO, USA) were dissolved in HPLC grade methanol. The absorbance wavelength used for the detection was as follows; ABA, GA, and CKat 254 nm, IAA at 280 nm. The LB medium without any inoculum was used as the blank.

### 2.7. Germination Experiment

The germination experiment was performed in the Plant Physiology Laboratory of PMAS Arid Agriculture University, Rawalpindi, Pakistan. Seeds of two wheat varieties, Pak 13 and NARC 09, were obtained from the National Agricultural Research Centre, Islamabad, Pakistan. Surface sterilization of seeds was done by 0.2% sodium hypochlorite for 5 min and later washed with distilled water. Drought tolerant bacterial strains, identified as *Bacillus subtilis* and *Azospirillum brasilense*, which also showed EPS secreting abilities, were used in this experiment. These bacterial strains and their combination were grown in LB broth in an incubator shaker for 48 h at 28 °C. After centrifugation at 4000 rpm for 10 min, the pellet was mixed with distilled water and optical density was adjusted at 1 (at 660 nm) for obtaining the final concentration of $10^8$ CFU/mL. Seeds were primed with *B. subtilis* and *A. brasilense* and their combination (as mentioned in Section 2.3) at 26 ± 2 °C for 4 h. Ten seeds were placed per petri plate and the treatments included; control, PEG (6000) 20% exposed seeds, and seeds primed with *B. subtilis*, *A. brasilense*, and the combination were placed in both normal and water-stressed conditions. Germination parameters like germination percentage, seedling vigor index (S.V.I), and promptness index (P.I) were measured.

### 2.8. Pot Experiment

A pot experiment was conducted in the glasshouse of PMAS Arid Agriculture University, Rawalpindi, Pakistan. Pots having 10 kg capacity were filled with soil and kept in the glasshouse. For each treatment, three replicates were taken and four plants were maintained per pot. Seeds of wheat varieties i.e., Pak 13 and NARC 09, were obtained from the National Agricultural Research Centre and sterilized by using 0.2% sodium hypochlorite. Seven seeds were planted per pot and thinning was done after germination to maintain four healthy plants per pot. The culture of bacterial isolates was used for the germination experiment (*B. subtilis*, *A. brasilense*, and combination, broth $10^8$ CFU/mL). Hundred mL of LB broth was used to grow the bacterial strains at 28 ± 2 °C for 48 h. The bacterial cells were harvested by centrifugation at 6000× *g* for 10 min and washed with phosphate-buffered saline (PBS). The final pellet was resuspended in PBS and cell density was adjusted to $10^8$ CFU/mL. The sterilized seeds were soaked into bacterial cell suspensions for 4 h. The uninoculated plant seeds soaked in sterile water only served as control. Uninocualted and inoculated seeds were sown in pots. Drought stress was induced after 1 month by withholding water and maintaining field capacity at 45%. After 15 days of imposing stress, different parameters i.e., morphological, physiological, and biochemical were analyzed.

### 2.9. Morphological Parameters

Morphological parameters like root length, shoot length, and leaf area were measured. Leaf area was measured with the help of a leaf area meter.

### 2.10. Physiological Parameters

#### 2.10.1. Membrane Stability Index Percentage (MSI%)

Leaves were collected and washed with distilled water. Then leaves were placed in test tubes having 10 mL distilled water and were kept in a water bath at 40 °C for 30 min, and electrical conductivity (C1) was recorded. Later, the same samples were placed in a water bath for 10 min at 100 °C and electrical conductivity (C2) was noted. The membrane stability index was calculated by using the formula [28].

$$\text{MSI\%} = [1 - C1/C2] \times 100 \tag{2}$$

#### 2.10.2. Estimation of Chlorophyll Content

Chlorophyll content was estimated by following the protocol of Bruinsama [29]. One gram of leaf was ground and homogenized in 80% acetone followed by centrifugation and measurement of absorbance at 470 nm, 663 nm, and 645 nm in UV-Visible spectrophotometer (Labomed UVD 3500, Los Angeles, CA, USA).

#### 2.10.3. Estimation of Water Potential and Osmotic Potential

The water potential of leaves was estimated by using a pressure chamber [30]. The osmotic potential was measured by using the protocol of Capell and Doerffling [31]. Leaves were placed in Eppendorf tubes and were frozen for 2 weeks. Afterward, the cell saps were collected and osmotic potential was recorded by using an osmometer.

### 2.11. Biochemical Parameters

#### 2.11.1. Estimation of Proline Content

Proline contents were determined by following the protocol of Bates [32]. The filtrate of fresh leaves (0.5 g) homogenized with 3.0% of sulfosalicylic acid (10 mL) was mixed with equal amounts of ninhydrin reagent (1.25 g ninhydrin in 30 mL glacial acetic acid and 20 mL 6 M phosphoric acid) and glacial acetic acid (100% pure, Sigma Aldrich). The mixture was heated in a water bath at 90 °C and the reaction was stopped after 1 h by transferring the mixture to ice. Toluene (1 mL) was added to the mixture and the resulting solution was separated into two distinct layers. Among these, the upper layer was used to determine the proline content by measuring the absorbance at 520 nm and using the proline standard curve.

#### 2.11.2. Estimation of Amino Acid Content

The extract was prepared for the analysis of amino acid content by following the method of Hamilton and Vanslyke [33]. Leaf extract (1 mL) was mixed with 1 mL of 80% ethanol, 1 mL of 0.2 M citrate buffer (pH-5), and 2 mL of the ninhydrin reagent. A spectrophotometer was used to measure the absorbance of the reaction mixture at 570 nm. The amino acids present in the samples were calculated using the following equation;

$$\text{Amino acids} = \frac{\text{Absorbance} \times \text{Volume} \times \text{Diluted concentration}}{\text{Weight of Plant Sample}} \times 1000$$

The standard curve was prepared by using the amino acid leucine and results were expressed in mg of amino acid per gram of dry tissue.

### 2.11.3. Estimation of Soluble Protein Content

Soluble protein content was estimated by Bradford assay [34]. Proteins were extracted by dissolving leaf samples (0.2 g) in 4 mL of sodium phosphate buffer (pH 7), and 0.5 mL of the extract was mixed with 3 mL of Coomassie bio red dye. Bovine serum albumin was used as a standard. Protein content was determined by measuring the optical density of the solution at 595 nm and using a protein standard curve.

### 2.11.4. Estimation of Soluble Sugar Content

Soluble sugar content was determined by using the protocol of Dubois et al. [35]. Ground plant tissue (0.1 g) was mixed with 80% methanol (3 mL). The solution was heated in a water bath at 70 °C for 30 min. An equal volume of 5% phenol and 0.5 mL extract was mixed with 1.5 mL of concentrated sulfuric acid and was again incubated in the dark for 30 min. The absorbance of the sample was checked at 490 nm. The standard curve for glucose solution was prepared which was used for the determination of the sugar content, expressed in mg/g/FW.

### 2.12. Estimation of Antioxidant Enzymes

Leaf (1 g) was ground in liquid nitrogen to get the enzyme extract. The obtained powder was added to 50 mM phosphate buffer (10 mL) at pH 7.0 and was then mixed with 1 mM ethylene diamine tetraacetic acid (EDTA) and 1% polyvinylpyrrolidone (PVP). The whole mixture was spun at 13,000× $g$ for 20 min at 4 °C. The resulting supernatant was used for the enzyme assay.

The SOD activity was measured by monitoring the inhibition in the photoreduction of nitro blue tetrazolium (NBT). The reaction mixture contained 130 mM methionine, 0.75 mM NBT, 0.05 M phosphate buffer (pH 7.0), 0.02 mM riboflavin and 300 µL enzyme extract. The reaction mixture and blank were exposed to fluorescent light for 7 min and absorbance was taken at 560 nm [36]. The CAT activity was determined by the change in absorbance due to $H_2O_2$ at 240 nm for 1 min [37]. The CAT activity (U/mg protein) was estimated from the molar absorption coefficient of 40 $mm^{-1}$ $cm^{-1}$ for $H_2O_2$. The POD activity was detected according to the method of Zhang and Karkim [38]. The reaction mixture consisted of 10 µL of crude enzyme extract, 10 µL of 100 mM $H_2O_2$, 160 µL of 50 mM sodium acetate (pH 5.0), and 20 µL of 100 mM guaiacol. Absorbance was recorded at 450 nm.

### 2.13. Statistical Analysis

Analysis of variance (ANOVA) was carried out to investigate the effects of drought stress on the accumulation of osmolytes and phytohormones in EPS producing *A. brasilense*, *B. subtilis*, and their combination. Similarly, differences in emulsification activity (%) and lyophilized chemical composition of EPS of *A. brasilense*, *B. subtilis*, and their combinations were analyzed through ANOVA. The effect of inoculation of EPS producing *A. brasilense*, *B. subtilis*, and their combinations in improving morphological (root and shoot length), physiological and biochemical attributes of two selected wheat varieties under control and drought stress conditions were also explored with ANOVA. The level selected for statistical significance was $p < 0.05$ and post-hoc comparison Tukey Honest Significant Difference (HSD) was used for mean separation following ANOVA [39]. All statistical computations were performed on Statistix 8.1.

## 3. Results

### 3.1. Soil Analysis and Bacterial Isolation

Collected soil samples were sandy loam in texture with an electrical conductivity range of 15–16 dS/m, pH (8.0), and low nutrient concentration (Supplementary Table S1). Isolated bacterial strains were characterized based on colony morphology and characters (Supplementary Table S2). The isolated colonies were white to creamy and exhibited rough margins in the LB medium.

### 3.2. Phenotypic and Molecular Characterization of the Isolates

These isolates were gram-negative and, vibroid or rod-shaped. All of the strains were able to grow up to 5% PEG; 90% of strains showed tolerance at 10%, 30%, and 15% PEG, while only two strains i.e., A07 and B05 were able to grow at 20% and 25% PEG (Supplementary Table S3). These two strains also showed positive results for phosphorous solubilization, hydrogen cyanide, and siderophore production (Supplementary Table S4). Initially, these two strains were identified based on C/N source utilization pattern (Supplementary Table S5).

Molecular characterization of both strains was carried out by amplification of the 1500-bp region of 16S rRNA and blast analysis of obtained nucleotides (Supplementary Table S6). The strain A07 (1400 nucleotides) showed a close similarity (98%) to *Azospirillum brasilense* (Accession No. CP012917.1), whereas strain B05 (with 1480 nucleotides) had a sequence similarity (99%) with *Bacillus subtilis* (Accession No.NC000964.3). The accession numbers of these two strains were obtained by NCBI and thus the isolated PGPR strains were identified as *A. brasilense* and *B. subtilis* respectively. The 16S rRNA gene sequences of these isolates were deposited in NCBI genebak with Accession No. MT742977 and MT742976 respectively.

### 3.3. Drought Tolerance and Osmolytes in Bacterial Strains

Drought tolerance of bacterial strains was done by using polyethylene glycol (PEG6000) (Table 1). Bacterial strains were subjected to two levels of osmotic stress i.e., 10% and 20% PEG. A high concentration of proline and sugar was observed at 20% PEG in combination i.e., 37.5% proline and 45.8% sugar as compared to control. *A. brasilense* culture synthesized more amounts of osmolytes under osmotic stress (28.5% proline and 34.57% sugar) as compared to *B. subtilis* (15.38% proline and 31.1% sugar).

**Table 1.** Drought tolerance and osmolytes production (in μg/mg) of exopolysaccharides (EPS)-producing bacterial strains under osmotic stress conditions.

| Bacterial Strains | Proline | | | Sugar | | |
|---|---|---|---|---|---|---|
| | Control | 10% PEG | 20% PEG | Control | 10% PEG | 20% PEG |
| *B. subtilis* | 2.6g ± 0.05 | 2.8f ± 0.07 | 3e ± 1.3 | 47.29i ± 3.2 | 57f ± 5.7 | 62d ± 5.8 |
| *A. brasilense* | 2.8f ± 0.01 | 3.4c ± 0.04 | 3.6b ± 0.06 | 52.7h ± 7.4 | 61e ± 3.8 | 71b ± 4.2 |
| Combination | 3.2d ± 0.08 | 3.7b ± 0.03 | 4.4a ± 0.04 | 54.1g ± 5.3 | 65c ± 2.9 | 79a ± 9.3 |

Mean values of an osmolyte (proline) sharing different letters are significantly different ($p < 0.05$) from each other and vice versa.

### 3.4. The EPS-Emulsification Activity of Bacterial Strains

Results regarding EPS-emulsification activity (Table 2) revealed that strain *Bacillus subtilis* showed 45% and 32% activity after 30 and 60 min, respectively. Whereas *A. brasilense* showed 42% and 30% EPS activities after 30 and 60 min, respectively. The combination, however, retained 51% and 33% emulsification activity after 30 and 60 min, respectively. The bacterial combination performed better than either of the single strains.

**Table 2.** The EPS-emulsification activity of *B. subtilis*, *A. brasilense*, and their combination are represented by mean ± standard deviation values.

| Bacterial Strains | Emulsifying Activity (%) | |
|---|---|---|
| | After 30 min Incubation | After 60 min Incubation |
| *B. subtilis* | 45b ± 2.4 | 32e ± 1.6 |
| *A. brasilense* | 42c ± 5.2 | 30f ± 1.8 |
| Combination | 51a ± 8.3 | 33d ± 1.3 |

Mean values carrying different letters are significantly different ($p < 0.05$) from each other.

### 3.5. Solubility and Chemical Composition of EPS

The lyophilized EPS were water-soluble but insoluble in acetone, benzene, chloroform, and ethanol. Results of the chemical composition of EPS showed that it mainly contained polysaccharides, protein, soluble sugar, and uronic acid content (Table 3). The uronic acid contents were less than protein and sugar in all EPS-producing strains.

**Table 3.** The EPS solubility and chemical composition of *B. subtilis*, *A. brasilense*, and their combination provided information of mean ± standard deviation values.

| Bacterial Strains | Polysaccharides (mg/g) | Protein (µg/g) | Soluble Sugar (µg/g) | Uronic Acid (µg/g) |
|---|---|---|---|---|
| *B. subtilis* | 2.56c ± 0.026 | 720.3c ± 36.4 | 6954b ± 12.7 | 1.1187b ± 0.06 |
| *A. brasilense* | 4.07b ± 0.14 | 740.2b ± 22.7 | 6954b ± 31.8 | 1.1187b ± 0.02 |
| Combination | 5.73a ± 0.43 | 771.5a ± 25.9 | 6976a ± 22.5 | 1.1236a ± 0.07 |

Mean values within a column having different letters are significantly different from each other and vice versa at $p < 0.05$.

### 3.6. Screening and Production of Phytohormones

Bacterial strains were analyzed for phytohormones (Table 4) and the synthesis of IAA, Ck, GA, and ABA in stress and non-stress conditions was observed. *A. brasilense* showed a higher concentration of hormones compared to *B. subtilis*, while combination showed maximum production. There was a decrease in IAA, GA, and CK by 61%, 49%, and 30% under stress but an increase of 27.29% in ABA concentration in osmotic stress (10% PEG) compared to non-stress conditions.

**Table 4.** Phytohormones level in *B. subtilis*, *A. brasilense*, and their combination in µg/mL.

| Bacterial Strain | IAA | | GA | | CK | | ABA | |
|---|---|---|---|---|---|---|---|---|
| | Control | 20% PEG | Control | 20% PEG | Control | 20% PEG | Control | 20% PEG |
| *B. subtilis* | 12.03c ± 0.26 | 3.79e ± 0.05 | 7.36d ± 0.015 | 4.71e ± 0.03 | 3.5c ± 0.09 | 1.9e ± 0.07 | 23.09e ± 0.52 | 33.59d ± 2.08 |
| *A. brasilense* | 23.15b ± 0.18 | 7.23d ± 0.13 | 18.74b ± 0.32 | 10.15c ± 0.21 | 4.6b ± 0.08 | 2.1e ± 0.08 | 36.13d ± 2.35 | 61.03c ± 3.19 |
| Combination | 40.23a ± 0.53 | 12.49c ± 0.08 | 27.06a ± 1.38 | 19.13b ± 0.13 | 5.1a ± 0.1 | 2.5d ± 0.08 | 83.11b ± 5.08 | 116.23a ± 8.15 |

Mean values of phytohormone sharing different letters are significantly different ($p < 0.05$) from each other.

### 3.7. Germination of Wheat

Results related to the germination experiment (Supplementary Figure S1) showed that under osmotic stress (PEG 20%) germination and germination related traits decreased. Seeds inoculated with bacterial strains showed a high germination percentage i.e., 18.24% and 15.78% under osmotic stress as compared to the control (un-inoculated seeds). Germination percentage (G%), seedling vigor index (SVI) and promptness index (PI) values were high in combination-treated seeds as compared to single strain inoculated seeds. According to these findings, the treatments followed the pattern i.e., combination > *A. brasilense* > *B. subtilis*.

### 3.8. Pot Experiment

Results of the pot experiment showed that drought significantly reduced plant growth, and altered morphological, physiological, and biochemical processes. It also indicated better performance of combination for imparting drought tolerance in wheat, as compared to *B. subtilis* and *A. brasilense* alone.

### 3.9. Morphological Responses of Wheat

A significant difference ($p < 0.05$) was recorded in shoot length (Figure 1A) and root length of wheat (Figure 1B) treated with bacterial strains as compared to control in irrigated and drought exposed plants. The plants treated with combination showed significantly better results when compared with single bacterial strain inoculation. The increase due to combination was 42.89% and 41.31% in shoot length while 29.79% and 27.22% in root length in both varieties under drought as compared to un-inoculated plants facing drought stress.

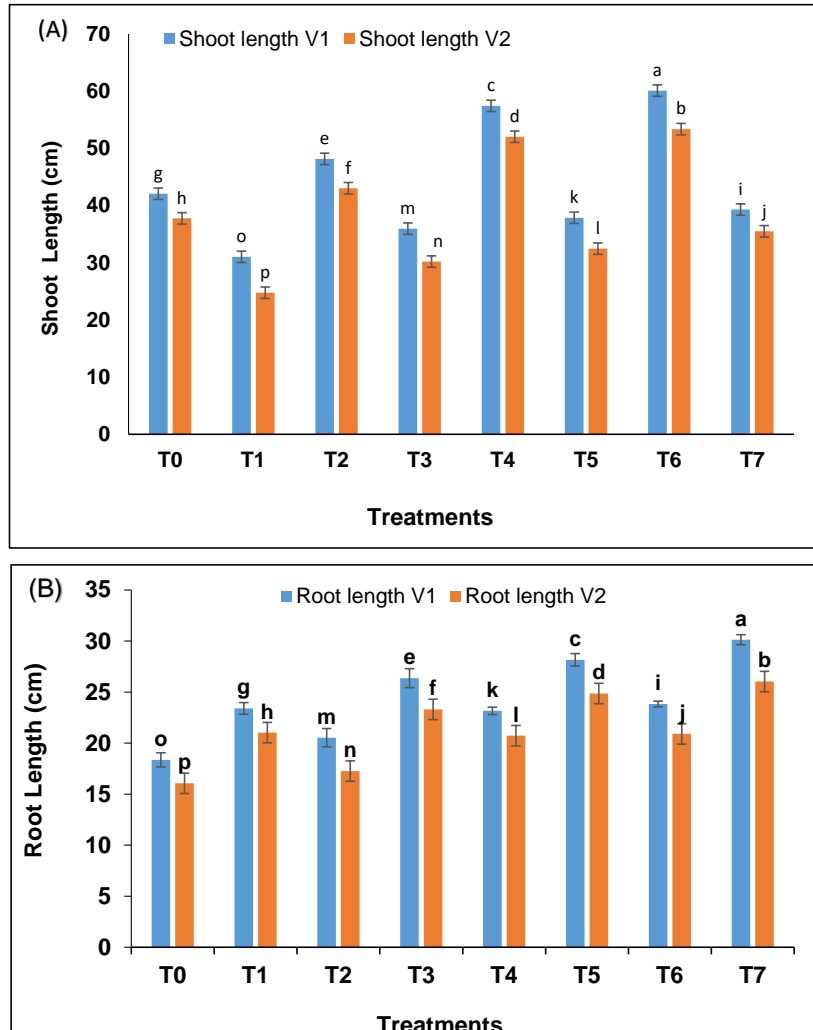

**Figure 1.** Effect of inoculation on (**A**) shootlength (cm) and (**B**) root length (cm) of two wheat varieties under control and droughtstress conditions. Where, T0 = control, T1 = drought stress, T2 = well-watered + *B. subtilis*, T3 = drought + *B. subtilis*, T4 = well-watered + *A. brasilense*, T5 = drought + *A. brasilense*, T6 = bacterial combination + well-watered, T7 = bacterial combination + drought, V1 = Pak 13 and V2 = NARC 09.Thecolumns anderror bars represent the mean values and standard deviations respectively whereas, different letters on mean values of either shoot or root length across seven treatments indicate significant difference at *p* < 0.05.

## 3.10. Physiological Responses of Wheat

Leaf area decreased in drought stress whereas, in well-watered plants, leaves were comparatively larger and exhibited enlarged surface area (Figure 2). Bacterial combination-treated plants showed greater leaf area (27.5 cm$^2$) followed by *A. brasilense* (24.96 cm$^2$) and *B. subtilis* (23.7 cm$^2$).

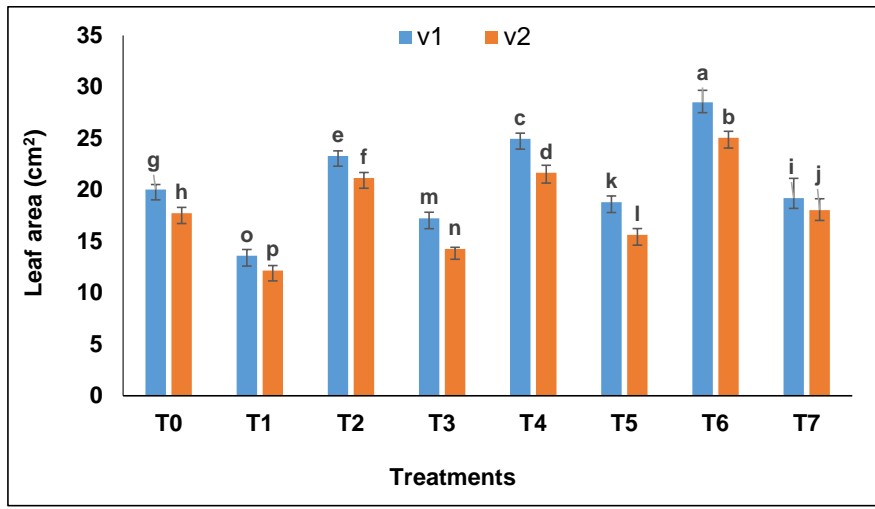

**Figure 2.** Effect of inoculation on leaf area (cm$^2$) of two wheat varieties under control and drought stress conditions. Where, T0 = control, T1 = drought stress, T2 = well-watered + *B. subtilis*, T3 = drought + *B. subtilis*, T4 = well-watered + *A. brasilense*, T5 = drought + *A. brasilense*, T6 = bacterial combination + well-watered, T7 = bacterial combination + drought, V1 = Pak 13 and V2 = NARC 09. The columns and error bars represent the mean values and standard deviations respectively whereas; different letters on mean values of leaf area indicate significant difference at *p* < 0.05 and vice versa.

The membrane stability index (Figure 3) of drought-exposed plants decreased significantly as compared to control i.e., 30.12% (Pak 13) and 34.61% (NARC 09). Bacterial strains increased membrane stability and the increase recorded was higher in Pak 13 as compared to NARC 09 i.e., 29.48% and 24.17%, respectively. The bacterial combination increased membrane stability efficiently as compared to single strains inoculation.

*A. brasilense* and *B. subtilis* inoculations also showed a positive response towards chlorophyll content (Table 5) as compared to control. Chlorophyll a and b content was higher in bacterial combination-treated plants in irrigated and drought exposed conditions. The increase in chlorophyll a, b and total chlorophyll content in combination-treated plants was 39.8%, 61.49%, and 45% as compared to control. Whereas *A. brasilense* and *B. subtilis* increased chlorophyll content by 31.14% and 18.62% respectively. The increase in chlorophyll content was higher in Pak 13 (45%) as compared to NARC 09 (42.44%).

Under drought, water potential was more negative i.e., −2.2 MPa as compared to control −0.5 Mpa (Figure 4A). Bacterial strains increased water potential in drought exposed plants to −1.4 MPa and −1.5 MPa. A similar trend was observed for osmotic potential (Figure 4B). Pak 13 showed a better response as compared to NARC 09.

**Table 5.** Effect of inoculation on chlorophyll and carotenoid contents of two wheat varieties under control and drought stress conditions.

| Treatments | Chlorophyll a (mg/g FW) | | Chlorophyll b (mg/g FW) | | Carotenoid (mg/g FW) | |
|---|---|---|---|---|---|---|
| | V1 | V2 | V1 | V2 | V1 | V2 |
| T0 | 5.2g ± 0.99 | 4.6h ± 0.84 | 3.61g ± 0.083 | 3.16h ± 0.13 | 1.51g ± 0.09 | 1.29h ± 0.12 |
| T1 | 1.87o ± 1.28 | 1.6p ± 1.15 | 1.52o ± 0.021 | 1.11p ± 0.04 | 0.79o ± 0.13 | 0.64p ± 0.18 |
| T2 | 5.9e ± 1.42 | 5.7f ± 1.29 | 4.86e ± 0.11 | 4.11f ± 0.11 | 1.96e ± 0.15 | 1.71f ± 0.13 |
| T3 | 2.96m ± 1.51 | 2.63n ± 1.21 | 2.71m ± 0.05 | 2.21n ± 0.03 | 1.28m ± 0.14 | 1.15n ± 0.11 |
| T4 | 6.46c ± 1.58 | 6.03d ± 1.38 | 5.13c ± 0.24 | 4.83d ± 0.12 | 2.2c ± 0.16 | 2.06d ± 0.14 |
| T5 | 3.13k ± 1.96 | 2.63l ± 1.71 | 3.19k ± 0.04 | 2.95l ± 0.06 | 1.42k ± 0.04 | 1.29l ± 0.16 |
| T6 | 7.13a ± 2.22 | 6.83b ± 2.06 | 6.1a ± 0.08 | 5.63b ± 0.13 | 2.93a ± 0.20 | 2.76b ± 0.17 |
| T7 | 4.6i ± 2.93 | 4.4j ± 2.74 | 3.71i ± 0.05 | 3.27j ± 0.09 | 1.58i ± 0.18 | 1.387j ± 0.19 |

Where, T0 = control, T1 = drought stress, T2 = well-watered + *B. subtilis*, T3 = drought + *B. subtilis*, T4 = well-watered + *A. brasilense*, T5 = drought + *A. brasilense*, T6 = bacterial combination + well-watered, T7 = bacterial combination + drought, V1 = Pak 13 and V2 = NARC 09. Thecolumns anderror bars represent the mean values and standard deviations respectively whereas; different letters on the mean values indicate significant difference at *p* < 0.05.

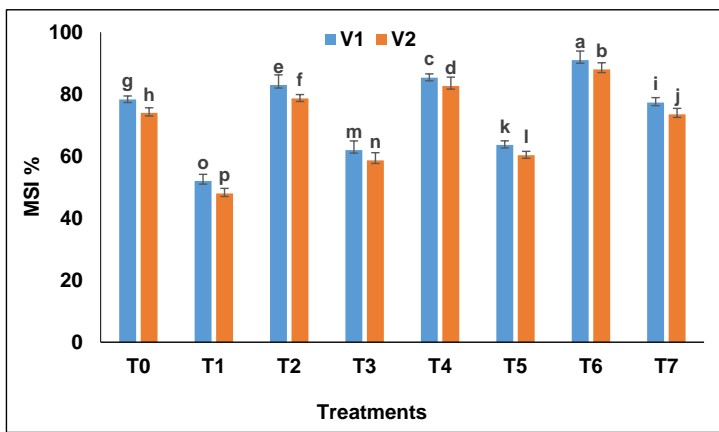

**Figure 3.** Effect of inoculation on membrane stability index (MSI) % of two wheat varieties under control and drought stress conditions. Where, T0 = control, T1 = drought stress, T2 = well-watered + *B. subtilis*, T3 = drought + *B. subtilis*, T4 = well-watered + *A. brasilense*, T5 = drought + *A. brasilense*, T6 = bacterial combination + well-watered, T7 = bacterial combination + drought, V1 = Pak 13 and V2 = NARC 09. The columns and error bars represent the mean values and standard deviations respectively whereas; different letters on mean values of MSI % indicate significant difference at $p < 0.05$ and vice versa.

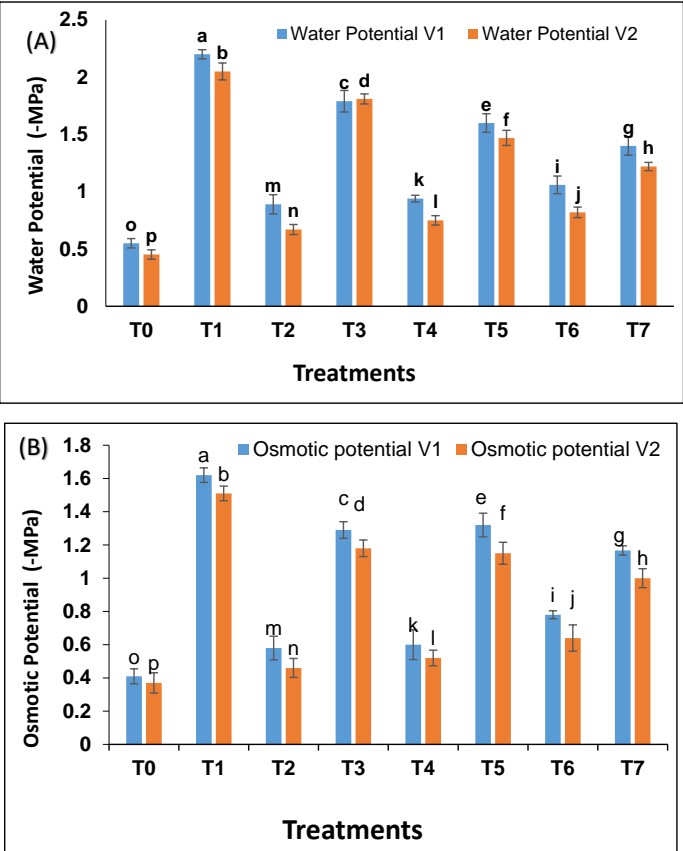

**Figure 4.** Effect of inoculation on (**A**) water potential and (**B**) osmotic potential of two wheat varieties under control and drought stressed conditions. Where, T0 = control, T1 = drought stress, T2 = well-watered + *B. subtilis*, T3 = drought + *B. subtilis*, T4 = well-watered + *A. brasilense*, T5 = drought + *A. brasilense*, T6 = bacterial combination + well-watered, T7 = bacterial combination + drought, V1 = Pak 13 and V2 = NARC 09. Thecolumns anderror bars represent the mean values and standard deviations respectively whereas; different letters on the mean values indicate significant difference at $p < 0.05$.

### 3.11. Biochemical Responses of Wheat

Bacterial strains significantly ($p < 0.05$) increased the concentration of the osmolytes i.e., proline, amino acids, protein, and sugar as compared to un-inoculated plants under drought stress. (Tables 6 and 7). The increase in proline content under drought was 14%, 28.12%, and 30% in *B. subtilis*, *A. brasilense*, and combination-treated plants respectively. The bacterial combination resulted in the maximum accumulation of osmolytes followed by *A. brasilense* and *B. subtilis*. A similar trend was observed for protein, sugar, and amino acid content in plants.

**Table 6.** Effect of inoculation on amino acid and proline content of two wheat varieties under control and drought stress conditions.

| Treatments | Amino Acid (mg/g FW) | | Proline (µg/g FW) | |
|---|---|---|---|---|
| | V1 | V2 | V1 | V2 |
| T0 | 0.25o ± 0.018 | 0.21p ± 0.029 | 32o ± 5.71 | 29p ± 2.94 |
| T1 | 0.69g ± 0.01 | 0.65h ± 0.013 | 106.66g ± 6.23 | 101.66h ± 1.89 |
| T2 | 0.35m ± 0.012 | 0.31n ± 0.015 | 45.33m ± 2.86 | 34n ± 6.23 |
| T3 | 0.75e ± 0.017 | 0.7f ± 0.017 | 135.66e ± 5.43 | 126.66f ± 2.94 |
| T4 | 0.41k ± 0.016 | 0.37l ± 0.019 | 56.33k ± 2.86 | 50l ± 4.78 |
| T5 | 0.82c ± 0.019 | 0.77d ± 0.014 | 151c ± 5.71 | 143.33d ± 5.09 |
| T6 | 0.49i ± 0.012 | 0.45j ± 0.012 | 71.33i ± 3.39 | 65.33j ± 4.64 |
| T7 | 0.9a ± 0.018 | 0.85b ± 0.016 | 164.66a ± 3.29 | 155.33b ± 2.86 |

Where, T0 = Control, T1 = Drought stress, T2 = Well-watered + *B. Subtilis*, T3 = Drought + *B. subtilis*, T4 = Well-watered + *A. brasilense*, T5 = Drought + *A. brasilense*, T6 = Bacterial combination + Well-watered, T7 = Bacterial combination + Drought, V1 = Pak 13, V2 = NARC 09, FW = Fresh weight. The data presented as mean ± standard deviation. Different letters with the mean values indicate significant difference at $p < 0.05$ and vice versa.

**Table 7.** Effect of inoculation on sugar and protein content of two wheat varieties under control and drought stress conditions.

| Treatments | Soluble Sugars (µg/g FW) | | Protein (µg/g FW) | |
|---|---|---|---|---|
| | V1 | V2 | V1 | V2 |
| T0 | 11.33o ± 1.24 | 10.04p ± 0.47 | 2.47o ± 0.14 | 2.43p ± 0.12 |
| T1 | 21.13g ± 0.47 | 19.48h ± 1.69 | 3.31g ± 0.01 | 3.18h ± 0.01 |
| T2 | 13m ± 1.63 | 12n ± 1.63 | 2.7m ± 0.21 | 2.6n ± 0.32 |
| T3 | 25e ± 0.81 | 23.41f ± 1.24 | 3.87e ± 0.2 | 3.7f ± 0.29 |
| T4 | 15.33k ± 1.24 | 14l ± 0.81 | 3k ± 0.37 | 2.83l ± 0.32 |
| T5 | 27.93c ± 0.89 | 26.12d ± 2.05 | 4.73c ± 0.12 | 4.36d ± 0.12 |
| T6 | 17i ± 0.81 | 16.66j ± 1.69 | 3.16i ± 0.26 | 3.13j ± 0.13 |
| T7 | 31.42a ± 1.24 | 29.37b ± 2.62 | 4.86a ± 0.16 | 4.53b ± 0.15 |

Where, T0 = Control, T1 = Drought stress, T2 = Well-watered + *B. Subtilis*, T3 = Drought + *B. subtilis*, T4 = Well-watered + *A. brasilense*, T5 = Drought + *A. brasilense*, T6 = Bacterial combination + Well-watered, T7 = Bacterial combination + Drought, V1 = Pak 13, V2 = NARC 09, FW = Fresh weight. The data presented as mean ± standard deviation. Different letters with the mean values indicate significant difference at $p < 0.05$ and vice versa.

Bacterial strains also increased the production of amino acids in wheat under irrigated and drought stress conditions. The amino acid content increased up to 6.84%, 12.32%, and 23.28% in inoculated plants as compared to un-inoculated drought exposed plants. The bacterial combination provided better results as compared to single strain inoculation. The increase in amino acid content was recorded high in Pak 13 (23.28%) as compared to NARC 09 (21.53%).

The increase in protein content was significant under drought stress as compared to control. Bacterial inoculation also increased protein synthesis in plants. Among inoculated plants, combination-treated plants showed better results with 46.82% and 42.45% increase as compared to untreated drought exposed plants. *A. brasilense* demonstrated better results as compared to *B. subtilis* in both wheat varieties.

Sugar production also increased under stress i.e., 26.18% and 21.44% as compared to control. Bacterial strains also increased the concentration of sugar, in wheat; the increase was 47.29%, 52.59%,

and 71.41% in Pak 13 and 45.94%, 50.37%, and 64.40% in NARC 09 in *B. subtilis*, *A. brasilense* and combination inoculations respectively compared to control.

### 3.12. Antioxidant Activities

Drought stress increased the activities of antioxidants (Table 8) as compared to control. The increase in SOD was 14.25%, 25.71%, and 35.71% in Pak 13 due to *B. subtilis*, *A. brasilense* and combination inoculation as compared to un-inoculated drought exposed plants whereas in NARC 09 the production increased by 10.29%, 22.05%, and 33.05%. It was noted that combination inoculation was more effective as compared to single strain inoculation.

**Table 8.** Effect of inoculation on antioxidants activity of two wheat varieties under control and drought stress conditions.

| Treatments | SOD (Units/g FW) | | CAT (Units/g FW) | | POD (Units/g FW) | |
|---|---|---|---|---|---|---|
| | V1 | V2 | V1 | V2 | V1 | V2 |
| T0 | 0.53n ± 0.04 | 0.43o ± 0.1 | 24.33m ± 1.2 | 22.57n ± 1.8 | 28.33o ± 1.6 | 24.66p ± 1.1 |
| T1 | 1.4g ± 0.08 | 1.36h ± 0.04 | 51.33d ± 1.5 | 43.57g ± 0.2 | 59g ± 1.2 | 48h ± 0.33 |
| T2 | 1.03l ± 0.09 | 0.93m ± 0.04 | 28l ± 0.9 | 26.96l ± 1.3 | 37m ± 0.8 | 31.33n ± 2.5 |
| T3 | 1.6e ± 0.07 | 1.5f ± 0.06 | 56.33c ± 1.4 | 47.11f ± 1.6 | 65.33e ± 1.2 | 62.66f ± 1.26 |
| T4 | 1.2j ± 0.06 | 1.16k ± 0.03 | 38.33i ± 2.16 | 34j ± 0.44 | 41k ± 1.1 | 39.41l ± 0.49 |
| T5 | 1.76c ± 0.03 | 1.66d ± 0.05 | 59b ± 1.47 | 49.33e ± 1.1 | 73.66c ± 1.2 | 69.63d ± 1.6 |
| T6 | 1.26i ± 0.05 | 1.2j ± 0.04 | 41.66h ± 0.1 | 31.33k ± 0.6 | 47.33i ± 0.6 | 43.25j ± 0.81 |
| T7 | 1.9a ± 0.06 | 1.81b ± 0.03 | 64.46a ± 0.6 | 57.03c ± 1.2 | 83a ± 0.94 | 77.51b ± 1.4 |

Where, T0 = Control, T1 = Drought stress, T2 = Well-watered + *B. Subtilis*, T3 = Drought + *B. subtilis*, T4 = Well-watered + *A. brasilense*, T5 = Drought + *A. brasilense*, T6 = Bacterial combination + Well-watered, T7 = Bacterial combination + Drought, V1 = Pak 13, V2 = NARC 09, FW = Fresh weight. The data presented as mean ± standard deviation. Different letters with the mean values indicate significant difference at $p < 0.05$ and vice versa.

The CAT production increased up to 41.28% in drought exposed plants as compared to control. Inoculation of *B. subtilis*, *A. brasilense*, and combination increased catalase activity up to 55.05%, 62.40%, and 77.42% in Pak13 and 29.89%, 58.66%, and 64.61% in NARC 09. However, Pak13 exhibited a higher level of antioxidants as compared to NARC 09.

Peroxidase production also increased for combating drought stress tolerance in wheat plants. Best results were obtained by combination inoculation in wheat i.e., 40.67% increase in Pak13 and 25.02% in NARC 09 was observed compared to un-inoculated plants under stress.

## 4. Discussion

Drought stress has negative effects on wheat germination and growth. It was reported that PGPR increased the drought tolerance of plants. They colonize the rhizosphere and induce drought tolerance through various mechanisms like changes in root architecture, siderophore production, osmoregulation, production of phytohormones, regulations of antioxidants, and most importantly, the production of large chain extracellular polysaccharide (EPS) which improves plant growth [40]. The PGPR strains, *B. subtilis* and *A. brasilense*, were screened for their drought tolerance ability. Their EPS secreting ability was also tested. It was noted that the concentration of EPS was higher in stress (PEG6000) culture as compared to control conditions. Both strains produced a significant amount of EPS in a culture facing osmotic stress. The EPS are hydrated compounds with 97% water in the matrix which protects the bacteria as well as plants from desiccation. A wide variety of PGPR have been reported to produce EPS and they help crop plants in better root colonization, good seed germination, and stress tolerance [41]. They enhance water retention by maintaining the diffusion of organic carbon sources [42]. They consist of polysaccharides, proteins, soluble sugars, and uronic acid. In response to osmotic stress, bacterial strains also synthesize proline and soluble sugar. Upadhyay et al. [43] and Vardharajula et al. [44] also reported such results. In this study, it was observed that both strains synthesized a high concentration of phytohormones like IAA, GA, and ABA. The IAA facilitates cell division, elongation, and differentiation while GA and ABA act as a signaling molecule in plants facing

stress conditions [45]. Synthesis of these hormones by bacterial strains imparts stress tolerance in plants [46].

A germination experiment was performed to check the ameliorative effect of selected bacterial strains. The germination parameters such as germination percentage, seedling vigor index, and promptness index decreased under high osmotic stress created by PEG 6000. Increased osmotic potential resulted in inhibition of water absorption by the seeds and impairment of other metabolic processes. In this study, bacterial combination increased germination and germination-related parameters significantly as compared to control. The EPS producing bacterial strains created a microenvironment that retained water and decreased osmotic stress. Under high osmotic stress, these stains synthesize extra amounts of EPS that alleviate the damage and increase the metabolic process in seeds [47]. Tewari and Arora [48] also reported a 50% increase in germination rate by the inoculation of EPS producing bacterial strains under stress.

Drought stress decreased plant growth as the essential nutrients and solutes are diverted to stress-related functions [49]. However, bacterial inoculation increased the growth of the plant. In this study, it was noted that root/shoot length and leaf area was increased significantly by the inoculation of bacterial strains as compared to control [50]. Aslam et al. [51] also reported that under drought, rhizobacteria increased the leaf area of plants. Mishra et al. [52] also reported an increase in the growth of plants under stress by the inoculation of rhizobacteria. These results were also in harmony with Mahmoud et al. [53] and Kumari et al. [54]. Due to the EPS secreting ability of these bacterial strains they easily colonize plants rhizosphere, adhere to the root surface, and maintain moisture content. They have adhesive properties and make stable aggregates that increase nutrients and water availability, which in turn improves plant development and growth [55]. It has been reported that plants inoculated with EPS producing bacterial strains are more drought tolerant because these strains maintain soil aggregate stability and hold water contents which in turn increase plant growth [56].

The results of the present study revealed that *B. subtilis* and *A. brasilense* increased osmotic potential, water potential, and chlorophyll content in plants. Danish et al. [57] also reported that under drought stress PGPR inoculated plants showed higher chlorophyll contents as compared to un-inoculated drought-exposed plants. The maintenance of water and osmotic potential under stress conditions is similar to the previous study by Khan et al. [58].

Osmolytes accumulation in a plant is the key indicator of drought tolerance as it performs osmotic adjustment and prevents water loss [59]. Major osmolytes are proline, sugar, and protein under water stress [60]. An increase in amino acid concentration is another factor that contributes to drought tolerance [61]. Our results showed that the amount of these osmolytes increased significantly under drought stress. It is one of the defense strategies of bacterial strains to make plants drought tolerant. In the current research, it was noted that the increase in osmolyte concentration was significant in inoculated plants and it was obvious that the bacterial strains contributed significantly to plant growth promotion under water scarcity by enhancing their defense strategies [62].

In normal conditions, the ROS are produced in plant cells in negligible amounts and are scavenged, but when the plant is facing stress, their concentration increases which damage DNA, lipids, cell membrane, and cellular organelles. In short, ROS cause cellular toxicity in plants growing under any stress. To reduce the damaging effects of ROS, plants synthesize antioxidants. Measuring the activities of antioxidant enzymes reveals the involvement of the scavenging system during drought and also indicates how much treatments were effective under stress [63]. In the current study, inoculated plants showed a high concentration of antioxidants (SOD, CAT, and POD) as compared to un-inoculated plants under stress. Pak-13 (V1) showed a fair improvement in antioxidant activities as compared to NARC-09 (V2). Our results are in harmony with Zahir et al. [64] and Rezazadeh et al. [65]. Gowtham et al. [66] also reported a significant increase in antioxidant enzymes by the inoculation of bacterial strains under drought stress conditions. These antioxidants have the potential to minimize oxidative damage and make plant drought tolerant [67].

## 5. Conclusions and Recommendations

Application of EPS producing bacterial strains in crops is an environment-friendly strategy which ameliorates the adverse effects of drought stress in plants. These strains use different mechanisms like osmolyte production, phytohormones production, and antioxidants synthesis which induce drought tolerance in plants. The bacterial combination was recorded better in improving morphological, physiological, and biochemical parameters of wheat under drought and it can be used as a potential inoculant in arid agro-ecologies.

**Supplementary Materials:** The following are available online at http://www.mdpi.com/2071-1050/12/21/8876/s1, Table S1: Physicochemical Characteristics of Soil Sample, Table S2: Morphology of isolates from the arid region of Pakistan, Table S3: Screening for drought tolerance of isolated strains, Table S4: Characteristics of isolated strains, Table S5: Carbon/Nitrogen source utilization pattern determined by QTS-24 kits, Table S6: Molecular identification of the isolates based on partial 16S rDNA analysis, Figure S1: Effect of inoculation on germination parameters.

**Author Contributions:** N.I.: conceptualization, writing-original draft, supervision; K.M.: investigation, writing-original draft; N.A.: data analysis, writing-original draft; H.Y.: formal analysis, writing-review and editing; R.Z.S.: writing-review and editing; W.K.: formal analysis, writing-review, and editing; H.A.E.E.: facilitation, review, proofreading; D.J.D.: formatting, review, proofreading; E.A.E.: funding acquisition, proofreading; Z.A.: statistical analysis, proofreading. All authors have read and agreed to the published version of the manuscript.

**Funding:** This research was funded in part by the UTM-RMC grant, Malaysia. The authors are also thankful for King Saud University, Riyadh, Saudi Arabia for funding through Researchers Supporting Project (Project No. RSP-2020/52).

**Conflicts of Interest:** The authors declare no conflict of interest.

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
