# Peer review of "Exopolysaccharides Producing Bacteria for the Amelioration of Drought Stress in Wheat"

_sustainability, doi:10.3390/su12218876_

Round 1
Reviewer 1 Report
I still find the article interesting, it has so much work and it is well planned, but the statistical analysis is a mess and it is still very difficult to figure out the kind of comparisons authors performed, specially in graphs.
You can see Bars with identical or very close values, and different letters, and very different values for other parameters in which letters are the same. I just cannot understand what kind of analysis has been performed there. The text still needs proofreading.
Without getting into details, this is the overall summary of my review.
Author Response
We are really thankful to the reviewer for the positive inputs and patience. In fact, combining various parameters resulted in all this vague look. We have removed all such graphs and have presented each parameter individually and moreover have also converted one complex graph into table
Hope it better depicts the results now.
Moreover, whole manuscript has been reviewed for typographical errors
Reviewer 2 Report
Previous Comments:
The need of the substantial use of microorganisms in agriculture is now increasing; therefore, the present manuscript is of importance and could be published prior to some minor revision. The experimental design was being carefully conducted and the conclusions are supported by the results.
Some minor concerns:
1) Some linguistic and grammatical mistakes that exist and need to be corrected.
2) I would not recommend addressing to the combination of the 2 strains as "consortium". I would use the word combination throughout the text.
3) A macroscopic image of the plants subjected to drought stress alone and plants with strains inoculation would be helpful.
The authors have addressed successfully my previous comments.
Author Response
RESPONSE TO REVIEWER’S COMMENTS
AUTHORS RESPONSE: The authors are really thankful for valuable suggestion and comments of editorial team which helped in the improvement of manuscript. Keeping in view the suggestions the manuscript has been improved and we tried to incorporate the changes accordingly.
ANNOTATED RESPONSE TO REVIEWER 1
The need of the substantial use of microorganisms in agriculture is now increasing; therefore, the present manuscript is of importance and could be published prior to some minor revision. The experimental design was being carefully conducted and the conclusions are supported by the results.
AUTHORS RESPONSE: The authors are highly appreciative of positive comments and suggestions. This helped us a lot to improve the scientific quality of our revised MS. The whole manuscript has been revised keenly in the light of suggestions proposed by reviewers.
Some minor concerns:
- Some linguistic and grammatical mistakes that exist and need to be corrected.
AUTHORS RESPONSE The suggested typological, grammatical, and technical correction has been incorporated in all sections of the manuscript.
- I would not recommend addressing to the combination of the 2 strains as "consortium". I would use the word combination throughout the text.
AUTHORS RESPONSE: We are thankful to the reviewer for the suggestion. Corrected Page 6; line 152; Page line207
- A macroscopic image of the plants subjected to drought stress alone and plants with strains inoculation would be helpful.
AUTHORS RESPONSE: The graphical abstract has been changed in the context of valuable suggestions.
Reviewer 3 Report
The present manuscript entitled: "Exopolysaccharides producing bacteria for the amelioration of drought stress in wheat" tells the story from isolation to applicaton of PGPR on wheat. Even though the research is well presentend and structured, the reviewer has found some points where the authors can imporve their manuscript.
Abstract:
- L37: ")" is missing
- Abstract could be shorter and better structured
- if you write numbers, please be consistent in number of digits you show. e.g. "xy.z%"
Introduction:
- L74: "Under water..."
M&M:
- Methods for soil physio-chemical parameters are missing
- L109: this sentence is confusing. are you saying you streaked the bacteria for pure cultures??
- L131: (and many times more in whole manuscript): if you cite a publication for the methods etc. please use "author et al., (year)"
- L144: did you develop this primers or are they from a publication?
- L224: uninoculated
- L230: of a leaf area meter
- L312: the isolates
- L318-324: why was the resulting sequence of different lenght? and why did you blast it two times with different results? please rephrase this section
- Why did you somethime use 10% and sometimes 20% PEG to induce osmotic stress? e.g. 3.6 vs. 3.7
- Result section is very long with many figures and tables. Maybe think of moving some to supplementary
Discussion & cunclusion:
- you always only mention the EPS producing capacity, however other mode of actions e.g. siderophore production, osmoprotectants etc. could also have an impact on the observed results. Do not neglect this.
Author Response
RESPONSE TO REVIEWER’S COMMENTS
AUTHORS RESPONSE: The authors are really thankful for the valuable suggestions and comments of editorial team which helped in the improvement of the manuscript. Keeping in view the suggestions the manuscript has been improved and we tried to incorporate the changes accordingly.
ANNOTATED RESPONSE TO REVIEWER 2
The present manuscript entitled: "Exopolysaccharides producing bacteria for the amelioration of drought stress in wheat" tells the story from isolation to applicaton of PGPR on wheat. Even though the research is well presentend and structured, the reviewer has found some points where the authors can imporve their manuscript.
AUTHORS RESPONSE: The authors are really thankful for valuable suggestion and comments of editorial team which helped in the improvement of the manuscript. Keeping in view the suggestions the manuscript has been improved and we tried to incorporate the changes accordingly.
Abstract:
- L37: ")" is missing
AUTHORS RESPONSE: Corrected, now at line 36
- Abstract could be shorter and better structured
AUTHORS RESPONSE Abstract has been shortened and improved (line 26-49).
- If you write numbers, please be consistent in number of digits you show. e.g. "xy.z%"
AUTHORS RESPONSE: Corrected and numbers of digits are consistent now at Line 38, 41, 44,46
Introduction:
- L74: "Under water..."
AUTHORS RESPONSE: Corrected at page 3; line 71
M&M:
- Methods for soil physio-chemical parameters are missing
AUTHORS RESPONSE: Mentioned in section 2.1. and in bibliography
- L109: this sentence is confusing. are you saying you streaked the bacteria for pure cultures??
AUTHORS RESPONSE: Corrected at Page 4; line 106-108
- L131: (and many times more in whole manuscript): if you cite a publication for the methods etc. please use "author et al., (year)"
AUTHORS RESPONSE: This was according to journal format so couldn’t correct it
- L144: did you develop this primers or are they from a publication?
AUTHORS RESPONSE: These are universal primers and used for PCR-amplification of bacterial DNA. They are thoroughly mentioned in literature.
- L224: uninoculated
AUTHORS RESPONSE: corrected (now at line 222)
- L230: of a leaf area meter
AUTHORS RESPONSE: Corrected (now at line 228)
L312: the isolates
AUTHORS RESPONSE: Corrected (now at line 309)
- L318-324: why was the resulting sequence of different lenght? and why did you blast it two times with different results? please rephrase this section
AUTHORS RESPONSE: This section has been rephrased (Page 9, Line 316-line 322). This process involves amplification and sequencing of 16S rRNA gene sequence which may result in bands of variable length. For both strains, initially the Blast gave resemblance to some specific accession numbers and later these strains were submitted to NCBI and obtained accession numbers of these strains have been mentioned.
- Why did you somethime use 10% and sometimes 20% PEG to induce osmotic stress? e.g. 3.6 vs. 3.7
- AUTHORS RESPONSE: We are thankful to reviewer for highlighting this discrepancy. The whole data was rechecked and mistake in 3.6 has been corrected. Now 20% PEG have been mentioned at both places.
- Result section is very long with many figures and tables. Maybe think of moving some to supplementary
AUTHORS RESPONSE: One table and one figure have been moved to supplementary file.
Discussion & conclusion:
- You always only mention the EPS producing capacity, however other mode of actions e.g. siderophore production, osmoprotectants etc. could also have an impact on the observed results. Do not neglect this.
AUTHORS RESPONSE: It has been incorporated now at page 15-16; line 495-498.
This manuscript is a resubmission of an earlier submission. The following is a list of the peer review reports and author responses from that submission.
Round 1
Reviewer 1 Report
This manuscript is about the study effect of two bacteria strains Bacillus subtilis and Azospirillum brasilense on wheat drought tolerance. The experiments are well designed and my only suggestion is since there are lots of grammar mistakes, the authors need to ask native English speaker to improve the language of this manuscript.
Author Response
ANNOTATED RESPONSE TO REVIEWER 1
|
No. |
COMMENT |
ACTION/JUSTIFICATION |
|
1. |
This manuscript is about the study effect of two bacteria strains Bacillus subtilis and Azospirillum brasilense on wheat drought tolerance |
The authors are very thankful to the reviewer for spending his valuable time to review this MS. Moreover, your positive comments are very appreciative for authors. |
|
2. |
The experiments are well designed, and my only suggestion is since there are lots of grammar mistakes, the authors need to ask native English speaker to improve the language of this manuscript. |
The whole manuscript has been revised for typographical and grammatical English mistakes. Besides using “Grammarly” software to edit the revised MS, our senior co-authors have carefully done proofreading of the revised MS to remove language errors. |
Reviewer 2 Report
The need of the substantial use of microorganisms in agriculture is now increasing; therefore, the present manuscript is of importance and could be published prior to some minor revision. The experimental design was being carefully conducted and the conclusions are supported by the results.
Some minor concerns:
1) Some linguistic and grammatical mistakes that exist and need to be corrected.
2) I would not recommend addressing to the combination of the 2 strains as "consortium". I would use the word combination throughout the text.
3) A macroscopic image of the plants subjected to drought stress alone and plants with strains inoculation would be helpful.
Author Response
ANNOTATED RESPONSE TO REVIEWER 2
|
No. |
COMMENT |
ACTION/JUSTIFICATION |
Page No./ Ln No. |
|
1 |
The need of the substantial use of microorganisms in agriculture is now increasing; therefore, the present manuscript is of importance and could be published prior to some minor revision. The experimental design was being carefully conducted and the conclusions are supported by the results. |
The authors are highly appreciative of positive comments and suggestions. This helped us a lot to improve the scientific quality of our revised MS. The whole manuscript has been revised keenly in the light of suggestions proposed by reviewers.
|
|
|
2 |
Some minor concerns: Some linguistic and grammatical mistakes that exist and need to be corrected. |
The suggested typological, grammatical, and technical correction has been incorporated in all sections of the manuscript.
|
|
|
3 |
I would not recommend addressing to the combination of the 2 strains as "consortium". I would use the word combination throughout the text. |
The whole manuscript has been revised keenly and the word ‘consortium” has been replaced with the word “combination”. |
Page 3; line 29,31,36,37 Pahe 5; line 148 Page 6; line 197,200, 202 Page 7; line 212 Page 9; line 314, 319, 325, 326 Page 10; line 328, 329, 336, 337, 340, 345, 346 Page 11, line 352, 354, 356, 362, 367, 372 Page 12, line 376, 380,381, 385, 390 Page 13; line 395, 396, 402, 409, 410, 414, 415 Page 14; line 420, 424, 425 Page 15; line 433, 437, 438, 439, 445, 446, 449 Page 16; line 454, 455, 456, 461, 462, 471 Page 17, line 482, 483, 486, 490, 491, 494, 498 Page 18; line 521 Page 19; line 569 |
|
4 |
A macroscopic image of the plants subjected to drought stress alone and plants with strains inoculation would be helpful. |
The graphical abstract has been changed in the context of valuable suggestions. |
Page 2; line 22 |
Reviewer 3 Report
The work "Exopolysaccharides Producing Bacteria for the Amelioration of Drought Stress in Wheat" characterizes EPS-producing bacterial isolates and assesses their contribution to drought tolerance in wheat. The work is well designed, the amount of work is impressive and the conclussions are clear. However, the text needs deep proofreading, the methods section requires more elaboration (a lot of important details are missing or poorly explained), the nature of comparisons as well as data representation in the results section are not clear and confusing.
This is a nice work, but it requires extensive edition before is acceptable for publication. In the following lines i summarize some other comments and suggestions about formal aspects of the manuscript that I recommend addressing before attempting a new submission:
- "Plant growth-promoting bacteria" is shortened as PGPB. The shortened form PGPR refers to plant growth-promoting rhizobacteria.
- The use of acronyms should be consistent throughout the text. PGPB, PGPR, EPS are used in their shortened and expanded forms throughout the text. Other terms such as EC are used in their abbreviated form although they only appear once.
- Line 76: "but the role of PGPR to augment the antioxidant still needs to be explored". This has already been demonstrated with, for example, strains of Pseudomonas.
- Include the composition or reference of Nitrogen free bromothymol (NFB) semisolid medium.
- There's an abuse of "xxx assay was performed according to xxx" in the methods section. At least a brief summary of the protocol is recommended, at least in assays directly linked to the main topic of the paper, such as EPS emulsification activity and quantification.
- The subheadings explaining the estimation of biochemical parameters lacks essential information such as Absorbance values, incubation temperatures, inconsistencies in solution composition (a solution cannot be composed of 60% compound A and 60% compound B), or missing information about reagent brands, etc.
- Lines 239 - 240: Describe the aspect of colonies...in which medium? based on the methods section authors used more than one.
- In general, I miss in the text reference to tables and/or figures.
- Table 2. The units are ug/mg. This is ug of the compound per mg of? supernatant? The nature of statistical comparisons here and in every table and figure in the manuscript is confusing. Comparisons and statistical methods need to be better addressed in the text. Figures mostly, but also tables, should be represented fitting the performed comparisons. One cannot tell if a letter o in each data sub-group has the same statistical meaning as a letter o in a different data set if it is not well explained and properly represented.
- In table 3, it looks like authors compared sample ODs with Emulsification activity, which is incorrect. Again, the nature of comparisons in every table and figure needs to be explained or performed properly.
- The bacterial consortium preparation and composition needs to be explained in the methods section.
- In general, in figures and tables, the sentence "Differences are signifficant (p < 0.05)" does not explain the use of letters. Please, explain in every case.
- Line 279: "EPS strains" is not a thing. Use EPS producing strains, instead.
- Figures, in general, are confusing, the naming of treatments not intuitive, and the use of bars and lines selected randomly in some cases (figs 7-9). Please, reelaborate all the figures based on the comparisons performed and use bars or lines consistently. I do highlight again the need for a better explanation of the comparisons performed. Letters in graphs make no sense in many cases, probably because there's no information about the factors compared in the statistical analysis.
- Figures must be independent from the text. A better explanation of each chart in the legend is recommended. Avoid referring a different figure legend to explain abbreviations. The idea is that one can have an idea of the experimental procedure, treatment, comparisons and result of the statistical analysis without going back to the main text or to other figures.
Author Response
ANNOTATED RESPONSE TO REVIEWER 3
|
No. |
COMMENT |
ACTION/JUSTIFICATION |
Page No. /Line No. |
|
|
The work "Exopolysaccharides Producing Bacteria for the Amelioration of Drought Stress in Wheat" characterizes EPS-producing bacterial isolates and assesses their contribution to drought tolerance in wheat. The work is well designed, the amount of work is impressive and the conclussions are clear. However, the text needs deep proofreading, the methods section requires more elaboration (a lot of important details are missing or poorly explained), the nature of comparisons as well as data representation in the results section are not clear and confusing. |
Authors are very appreciative to the worthy reviewer. The whole manuscript has been thoroughly revised and improved according to valuable suggestions.
|
|
|
1. |
This is a nice work, but it requires extensive edition before is acceptable for publication. In the following lines i summarize some other comments and suggestions about formal aspects of the manuscript that I recommend addressing before attempting a new submission: |
The authors are thankful to the valuable comments and suggestions. This helped us a lot to improve the scientific quality of our revised MS. The whole manuscript has been revised keenly in the light of suggestions proposed by reviewers. The pointwise response to each of your comment/suggestions is given below. |
|
|
2. |
"Plant growth-promoting bacteria" is shortened as PGPB. The shortened form PGPR refers to plant growth-promoting rhizobacteria. |
The whole manuscript has been checked and PGPB has been replaced, if written |
|
|
3. |
The use of acronyms should be consistent throughout the text. PGPB, PGPR, EPS are used in their shortened and expanded forms throughout the text. Other terms such as EC are used in their abbreviated form although they only appear once. |
Whole manuscript has been rechecked for PGPB, PGPR and EPS and all discrepancies have been removed EC has been fully written |
Page 3; line 66 Page 4; line 69,71,72 77, 82, 85, 87, 92 Page 5, line 155, 156, 158, 162 Page 6; line 170, 171,172, 173, 176, 196 Page 9; line 308, 318, 321, 322, 333 Page 10, line 327, 330, 331, 334, 335 Page 12, line 376, 390 Page 17; line 502, 503, 504 Page 18; line 506, 507, 508, 522, 524, 526, 533, 537, 542, line 498 Page 19, line 566
|
|
4. |
Line 76: "but the role of PGPR to augment the antioxidant still needs to be explored". This has already been demonstrated with, for example, strains of Pseudomonas. |
This sentence has been re-written. |
Page 4; line 77-78 |
|
5. |
Include the composition or reference of Nitrogen free bromothymol (NFB) semisolid medium. |
Detailed composition of has Nitrogen free bromothymol (NFB) semisolid medium has been provided |
Page 4; line 101-106 |
|
6. |
There's an abuse of "xxx assay was performed according to xxx" in the methods section. At least a brief summary of the protocol is recommended, at least in assays directly linked to the main topic of the paper, such as EPS emulsification activity and quantification. |
Detailed procedures have been described in the section 2.4. EPS Emulsification Activity of Bacterial Strains 2.5. Quantification of EPS Substances
|
Page 5; line 155-160
Page 6; line 165-177 |
|
7. |
The subheadings explaining the estimation of biochemical parameters lacks essential information such as Absorbance values, incubation temperatures, inconsistencies in solution composition (a solution cannot be composed of 60% compound A and 60% compound B), or missing information about reagent brands, etc |
Detailed procedures have been provided for the estimation of all procedures in biochemical parameters. |
Page 7-8; Line 237-285 |
|
8. |
Lines 239 - 240: Describe the aspect of colonies...in which medium? based on the methods section authors used more than one. |
Name of the medium has been provided in both materials and methods and results section |
Page 4;line 108 Page 9; line 295 |
|
9. |
In general, I miss in the text reference to tables and/or figures. |
All tables and figures have been mentioned in the text of the result section. |
|
|
10. |
Table 2. The units are ug/mg. This is ug of the compound per mg of? supernatant? The nature of statistical comparisons here and in every table and figure in the manuscript is confusing. Comparisons and statistical methods need to be better addressed in the text. Figures mostly, but also tables, should be represented fitting the performed comparisons. One cannot tell if a letter o in each data sub-group has the same statistical meaning as a letter o in a different data set if it is not well explained and properly represented. |
Proline (μg mg-1 |
|
|
11. |
In table 3, it looks like authors compared sample ODs with Emulsification activity, which is incorrect. Again, the nature of comparisons in every table and figure needs to be explained or performed properly. |
ODs values have been removed from table 3 and only emulsification activity has been provided. |
Page 10; line 328 |
|
12. |
The bacterial consortium preparation and composition needs to be explained in the methods section. |
The bacterial consortium preparation and composition has been provided in section 2.3 |
Page 5. Line 148-149 |
|
13. |
In general, in figures and tables, the sentence "Differences are signifficant (p < 0.05)" does not explain the use of letters. Please, explain in every case. |
This statement has been explained in all tables and figures. |
|
|
14. |
Line 279: "EPS strains" is not a thing. Use EPS producing strains, instead. |
Thanks for highlighting this error. This discrepancy has been removed. |
Page 9; line 334 |
|
|
Figures, in general, are confusing, the naming of treatments not intuitive, and the use of bars and lines selected randomly in some cases (figs 7-9). Please, reelaborate all the figures based on the comparisons performed and use bars or lines consistently. I do highlight again the need for a better explanation of the comparisons performed. Letters in graphs make no sense in many cases, probably because there's no information about the factors compared in the statistical analysis. |
An effort to present multiple parameters in one figure created this ambiguity. Figures 7-9 have been replaced by tables 7-9 and all necessary information has been provided |
Page 15-17 |
|
15. |
Figures must be independent from the text. A better explanation of each chart in the legend is recommended. Avoid referring a different figure legend to explain abbreviations. The idea is that one can have an idea of the experimental procedure, treatment, comparisons and result of the statistical analysis without going back to the main text or to other figures. |
Legends of all figures have been changed and detailed description has been added for each figure. |
Whole results section |
Round 2
Reviewer 3 Report
Although authors have included many changes in the new version of the manuscript, many of them were not required or necessary. Conversely, most of the changes regarding the main issues I pointed out in the previous revision have not been adressed. The new version of the manuscript requires extensive proofreading. Most of the biochemical tests lack standard curves, there are crucial elements in the methods section poorly explained, and the nature of comparisons in each assay is still confusing and not explained at all. Additionally, the article includes 100 citations, which is excessive for this kind of publication.
Some comments and suggestions:
Line 66: plant growth-promoting bacteria (PGPR), change by plant growth-promoting bacteria (PGPB) or plant growth-promoting rhizobacteria (PGPR)
line 101: (3000 rpmx10 min) for 10min, remove 10 min from one of the two places
Line 149: equal volumes of separate cultures of both strains. This still doesn't explain how the bacteria combination was prepared. Bacterial concentrations? Resuspended in what? Do these 2 species reach the same concentrations in stationary phase? Explain well how inocula were prepared.
Lines 156, 157, 163, 174: 0.5, units?; heat, at which temperature; hexadecane (0.5 mL), which concentration?; some time, minutes, hours, days?
Lines 176-177 and throughout the manuscript: The EPS content in medium without inoculum was used to normalize the data. This cannot be used to normalize data. You mean, used as blank?
line 212: (B.subtilis, A. brasilense, and combination, broth 108 CFU/ml), does it mean you inoculated cells suspended in culture medium? If that is the case, it is not the best way to do it. Specify.
Line 213: inoculated in the rhizosphere. How? surface watering? pot immersion? which volume of inoculum?
Line 239: glacial nihydrin reagent, glacial? which concentration and brand?
Line 243 and throughout the paper: K value, explain. Without standard curves using known concentrations one cannot quantify compound concentrations.
Line 247: Volumes still confusing, what happens with citrate buffer? also 1 mL?
Line 257: Comassive, you mean Coomassie, I assume.
The nature of statistical comparisons is still missing. The meaning of letters in tables and figures needs to be explained. Some figure legends are in the main text. Many figure legends lack crucial information about treatments, factors included, etc. The name of treatments is confusing and no effort has been made in renaming and reorganizing data. Substituting the last figures by tables does not change the fact they are still confusing and the compared treatments have not been explained.
I acknowledge the good experimental dessign, the structure of the manuscript and I can tell the amount of work in this study is massive, but the manuscript requires deep reelaboration and cannot be published in its current form.